# The impact of the early COVID-19 pandemic on healthcare system resource use and costs in two provinces in Canada: An interrupted time series analysis

Seraphine Zeitouny[1,2‡], Douglas C. Cheung[3,4‡]*, Karen E. Bremner[3]*, Reka E. Pataky[1,5], Priscila Pequeno[6], John Matelski[7], Stuart Peacock[1,8], M. Elisabeth Del Giudice[9,10], Lauren Lapointe-Shaw[6,11,12], George Tomlinson[7], Andrew B. Mendlowitz[3,13,14], Carol Mulder[15,16], Teresa C. O. Tsui[6,17,18,19], Nathan Perlis[4,20], Jennifer D. Walker[6,21], Beate Sander[3,6,11,14], William W. L. Wong[6,22], Murray D. Krahn[3,6,11,14†], Girish S. Kulkarni[4,6,20]

1 Canadian Centre for Applied Research in Cancer Control, BC Cancer, Vancouver, British Columbia, Canada, 2 Centre for Health Services and Policy Research, School of Population and Public Health, University of British Columbia, Vancouver, British Columbia, Canada, 3 Toronto Health Economics and Technology Assessment (THETA) Collaborative, University Health Network, Toronto, Ontario, Canada, 4 Divisions of Urology and Surgical Oncology, Department of Surgery, Temerty Faculty of Medicine, University of Toronto, Toronto, Ontario, Canada, 5 School of Population and Public Health, University of British Columbia, Vancouver, British Columbia, Canada, 6 ICES, Toronto, Ontario, Canada, 7 Biostatistics Research Unit, University Health Network, Toronto, Ontario, Canada, 8 Faculty of Health Sciences, Simon Fraser University, Burnaby, British Columbia, Canada, 9 Sunnybrook Health Sciences Centre, Toronto, Ontario, Canada, 10 Department of Family and Community Medicine, Temerty Faculty of Medicine, University of Toronto, Toronto, Ontario, Canada, 11 Toronto General Hospital Research Institute, University Health Network, Toronto, Ontario, Canada, 12 Department of General Internal Medicine, Toronto General Hospital, Toronto, Ontario, Canada, 13 Toronto Centre for Liver Disease/Viral Hepatitis Care Network (VIRCAN), University Health Network, Toronto, Ontario, Canada, 14 Institute of Health Policy Management and Evaluation, University of Toronto, Toronto, Ontario, Canada, 15 Chiefs of Ontario, Toronto, Ontario, Canada, 16 Queen's University, Kingston, Ontario, Canada, 17 Child Health and Evaluative Sciences, Hospital for Sick Children, Toronto, Ontario, Canada, 18 Sunnybrook Research Institute, Toronto, Ontario, Canada, 19 Canadian Centre for Applied Research in Cancer Control, Ontario Health-Cancer Care Ontario, Toronto, Ontario, Canada, 20 Division of Urology, Sprott Department of Surgery, University Health Network, Cancer Clinical Research Unit, Princess Margaret Cancer Centre, Toronto, Ontario, Canada, 21 Department of Health Research Methods, Evidence and Impact, McMaster University, Hamilton, Ontario, Canada, 22 School of Pharmacy, University of Waterloo, Kitchener, Ontario, Canada

† Deceased.

‡ SZ and DCC share joint first authorship on this work.

* douglas.cheung@mail.utoronto.ca (DCC); karen.bremner@uhnresearch.ca (KEB)

**Data Availability Statement:** In Ontario, the dataset from this study is held securely in coded form at ICES. While legal data sharing agreements

## Abstract

### Introduction

The aim of our study was to assess the initial impact of COVID-19 on total publicly-funded direct healthcare costs and health services use in two Canadian provinces, Ontario and British Columbia (BC).

### Methods

This retrospective repeated cross-sectional study used population-based administrative datasets, linked within each province, from January 1, 2018 to December 27, 2020.

between ICES and data providers (e.g., healthcare organizations and government) prohibit ICES from making the dataset publicly available, access may be granted to those who meet pre-specified criteria for confidential access, available at www.ices.on.ca/DAS. The full dataset creation plan and underlying analytic code are available from the authors upon request, understanding that the computer program may rely on coding templates or macros that are unique to ICES and therefore either inaccessible or may require modification. In British Columbia, access to data provided by the Data Steward(s) is subject to approval, but can be requested for research projects through the Data Steward(s) or their designated service providers.

**Funding:** Funding/support This study was funded by a Canadian Institutes of Health Research (CIHR) operating grant: COVID-19 Rapid Research Funding Opportunity (funding reference number VR4 172774) to Dr. Krahn and Dr. Kulkarni. This study was supported by ICES, which is funded by an annual grant from the Ontario Ministry of Health (MOH) and the Ministry of Long-Term Care (MLTC). This work was also undertaken, in part, thanks to funding from the Canada Research Chairs program to Dr. Krahn, Dr. Walker and Dr. Sander and an Ontario Early Researcher Award to Dr. Wong. Dr. Mendlowitz has received postdoctoral fellowships from the Canadian Network on Hepatitis C (CanHepC) and the Canadian Institutes of Health Research. CanHepC is funded by a joint initiative of the Canadian Institutes of Health Research (NPC-178912) and the Public Health Agency of Canada. The Canadian Centre for Applied Research in Cancer Control is funded by the Canadian Cancer Society.

**Competing interests:** The authors have declared that no competing interests exist.

Interrupted time series analysis was used to estimate changes in the level and trends of weekly resource use and costs, with March 16–22, 2020 as the first pandemic week. Also, in each week of 2020, we identified cases with their first positive SARS-CoV-2 test and estimated their healthcare costs until death or December 27, 2020.

## Results

The resources with the largest level declines (95% confidence interval) in use in the first pandemic week compared to the previous week were physician services [Ontario: -43% (-49%,-37%); BC: -24% (-30%,-19%) (both p<0.001)] and emergency department visits [Ontario: -41% (-47%,-35%); BC: -29% (-35%,-23%) (both p<0.001)]. Hospital admissions declined by 27% (-32%,-23%) in Ontario and 21% (-26%,-16%) in BC (both p<0.001). Resource use subsequently rose but did not return to pre-pandemic levels. Only home care and dialysis clinic visits did not significantly decrease compared to pre-pandemic. Costs for COVID-19 cases represented 1.3% and 0.7% of total direct healthcare costs in 2020 in Ontario and BC, respectively.

## Conclusions

Reduced utilization of healthcare services in the overall population outweighed utilization by COVID-19 patients in 2020. Meeting the needs of all patients across all services is essential to maintain resilient healthcare systems.

## Introduction

Healthcare news coverage in the latter part of 2022 was dominated by stories of a system in crisis [1–3]. Worldwide, the COVID-19 pandemic challenged the resilience of health systems, and revealed their strengths and limitations [4]. Analysis of the broad effects of the shock of the COVID-19 pandemic on healthcare services can help health systems prepare for future crises. While Canadian healthcare systems strived to meet the needs of COVID-19 patients, their immediate "ramp-down" strategies [5, 6] created backlogs in non-COVID-19 care [7] and potentially compromised the healthcare, and possibly the health, of many patients [6, 8–11]. In Ontario and British Columbia (BC), two provinces in Canada, non-COVID-19 healthcare services were alternately reduced and restarted as COVID-19 case counts fluctuated in communities [12, 13].

The initial ramp-down started in March 2020 when Canadian healthcare systems began reducing elective procedures and non-emergency care [5, 6, 8], and reconfiguring facilities in preparation for COVID-19 patients [14, 15]. In Canada and internationally, there were reports of major reductions in cancer screening [16–19], diagnosis [20], and treatment [21, 22] in 2020 compared to previous years, as well as lower numbers of hospital admissions [23, 24] and emergency department (ED) visits [10, 11, 25–27]. While some non-COVID-19 care was transitioned from in-person to virtual [28–32] and from inpatient to outpatient [33], concerns about delivering ideal care under COVID-19 circumstances were identified [34–36]. There is a large body of literature focussing on individual disease groups or resource categories, but limited studies have evaluated the costs and healthcare resource use of the entire health system.

Thus, the aim of our study was to assess the initial impact of COVID-19 directives and related mitigation measures on total healthcare costs and resource use across inpatient, outpatient, and long-term care in a concerted fashion for the populations in two provinces within

the Canadian universal health system. This broad scope of healthcare services addresses an important information gap by allowing comparisons across resources for a holistic understanding of how the healthcare system reacted and which resources were prioritized within the pandemic response. Specifically, we assessed the changes in the trends of publicly-funded healthcare services use and direct healthcare costs in Ontario and BC.

## Methods

### Data sources and settings

Our study analyzed healthcare utilization and costs using healthcare administrative datasets that capture publicly-funded outpatient care, inpatient hospital and long-term care, home care services, and prescription drugs for nearly all residents in Ontario and BC. In both provinces, data are collected for administrative purposes by the provincial Ministries of Health, and under the provincial universal health insurance plans [37, 38]. The datasets for Ontario are housed at ICES [38]. These datasets were linked using unique encoded identifiers and analyzed at ICES. Data from the BC Ministry of Health [39–46] were provided through Population Data BC [47]. These datasets were linked using unique encoded identifiers and analyzed by the BC research team. A description of the data sources is in S1 Table in S1 File.

This study was approved by the Research Ethics Boards at the University Health Network in Toronto, Ontario and BC Cancer. ICES is an independent, non-profit research institute whose legal status under Ontario's health information privacy law allows it to collect and analyze healthcare and demographic data, without consent, for health system evaluation and improvement. In BC, a waiver of consent was granted due to the nature of the data and minimal risk of the study. Guidance on Reporting of studies Conducted using Observational Routinely-collected health Data (RECORD) statement was followed (S1 Checklist in S1 File) [48].

**Study design and inclusion criteria.** This was a repeated cross-sectional study from a health system perspective. We examined repeated weekly patient-level measures of healthcare costs and resource use using a micro-costing approach from January 1, 2018, to December 27, 2020, in Ontario and BC separately. In each weekly time interval, defined according to the calendar of the International Organization of Standardization (ISO) [49–51], the study population included all individuals who were alive and registered in the provincial public health plan (Ontario Health Insurance Plan [52] (OHIP) and BC Medical Services Plan [45] (MSP), respectively) for at least one day.

In order to estimate the costs incurred by COVID-19 patients, we identified people with their first positive SARS-CoV-2 test result in each week of 2020 from the Ontario C19INTGR COVID-19 database [53, 54] and the BC Ministry of Health COVID-19 Test Lab Data [46, 55] and followed them until death or December 27, 2020, over two time periods: acute COVID-19, defined as the first 4 weeks (inclusive) after the positive test, and post-acute COVID-19, from the end of the acute COVID-19 period to end of observation [56, 57]. Individuals were in the non-COVID-19 group as long as they did not have a positive SARS-CoV-2 test result.

### Study outcomes and statistical analysis

Population characteristics of interest included age, sex, rural/urban residence [58, 59], neighbourhood income quintile, Ontario Marginalization Index quintiles [60, 61] (Ontario only), Ontario Local Health Integration Network (LHIN) [62] or BC Health Authority [63] of residence, and a measure of comorbidity derived from the weighted Aggregated Diagnosis Groups (ADGs) from John's Hopkins' Adjusted Clinical Groups® System (ACG®) version 10 [64, 65]. This score, developed using administrative data from Ontario, represents the relative risk of one-year all-cause mortality [64]. Population characteristics as of January 1, 2020 were

summarized using means and standard deviations for continuous variables, and frequencies and percentages for categorical variables.

Databases and healthcare services used in our analyses are described in S2 Table in S1 File.

For our analyses of healthcare resource use in each weekly interval, we calculated rates per 10,000 person weeks of acute hospital admissions, emergency department visits, physicians' services, same day surgery, outpatient prescription drug claims (restricted to prescription drugs covered under the provincial drug benefit plan for eligible individuals in Ontario [66], but including all prescription drugs dispensed in BC irrespective of the insurance payer), laboratory tests including SARS-CoV-2 tests, home care services, and admissions to publicly-funded long-term care (LTC) and complex continuing care (CCC) facilities. Rates of visits to cancer clinics and dialysis clinics were included for Ontario only, as these were not made available in BC. We used standard established methods for estimating healthcare costs from administrative data [67, 68] and we computed the cost of health services used by active registered residents, including hospitalization, emergency department visits, same day surgery, outpatient clinics (Ontario only), physicians' services, laboratory tests, home and community care, LTC, CCC, and outpatient prescription medications. Costs for all years were expressed in 2020 Canadian dollars (CAD) using Statistics Canada's Consumer Price Index for healthcare [69].

We generated descriptive analyses of healthcare service use, absolute cost of healthcare services, and rates of utilization and costs per 10,000 person-weeks in each week from 2018 to 2020, in the full population.

The first analysis examined healthcare costs and utilization of select healthcare resources, namely hospital admissions, emergency department visits, physicians' services, dialysis clinics (Ontario only) and cancer clinics (Ontario only), and home and community care in the full population. Interrupted time series (ITS) analysis was used to estimate changes in the level and trends of the study outcomes, with March 16–22, 2020, when COVID-19 was declared a pandemic in Canada, as the first week of the pandemic [5, 70–72]. ITS models were fit using linear mixed effects regression (LMER) using a Gaussian likelihood on the weekly outcome rates. Visual inspection of the weekly outcome data over the study period informed the specification of the fixed and random effects: for the fixed effects, a linear time trend in study week was estimated for the pre-period, a wild-point indicator was used for the first week of the pandemic, and a quadratic function in week number was used in the pandemic period. Indicator variables for the last week of December, the first week of January, and weeks with other statutory holidays were also included as fixed effects. Each of the above terms was allowed to interact with an indicator variable for province, which afforded separate estimates for each province while fitting only one generalized linear mixed model per outcome. A random intercept for each week of the ISO calendar (i.e., week number within year) was specified as a means of modelling shared seasonal effects and reducing residual autocorrelation [73]. We examined autocorrelation plots both with and without inclusion of the lag-1 outcome as a predictor and determined that its inclusion was generally helpful in ensuring acceptable levels of residual autocorrelation for the final model fits. Alpha of 0.05 was adopted as the threshold for statistical significance. Analyses were conducted using SAS version 9.4 and R version 3.6.1.

A second analysis compared the use of each resource in each week of 2020 to the average weekly use of the resource over both 2018 and 2019, and plotted the percent change in each week of 2020.

A third analysis tabulated healthcare cost estimates in each quarter (consecutive 13 weeks) of 2020 for the full population. The percent change in cost for each resource in each quarter of 2020 compared to its pre-pandemic level was calculated.

A fourth analysis calculated the percentage of the absolute healthcare costs incurred by individuals who tested positive for COVID-19 in 2020 in each province, and segregated by

duration since COVID-19 infection, relative to the direct healthcare costs incurred by the full population in 2020.

## Results

### Description of study populations

In the first week of 2020, the study populations included 14,973,016 individuals in Ontario and 5,123,554 in BC (Table 1). In Ontario, the mean (standard deviation) age was 41.2 (23.0) years, 50.6% of individuals were female, 89.7% of the population resided in urban areas, and 89.1% had a weighted ADG score between -19 and 20. The BC study population had a mean age of 42.2 (23.0) years, 50.6% were female, 88.8% resided in urban areas, and 86.9% had an ADG score between -19 and 20.

### Resource use by category

The estimated coefficients of the ITS model are presented in S3 to S6 Tables in S1 File. At the declaration of the pandemic in Canada (March 16 to 22, 2020) as compared to the last pre-pandemic week (March 9 to 15, 2020), the largest level declines (95% confidence interval) in resource use were physician service claims [Ontario: -43% (-49% to -37%), $p<0.001$; BC: -24% (-30% to -19%), $p<0.001$] and ED visits [Ontario: -41% (-47 to -35%), $p <0.001$; BC: -29% (-35% to -23%), $p<0.001$] (Fig 1a and 1b). These corresponded to immediate level drops of 1,437 (1,259 to 1,616) physician service claims per 10,000 person-weeks in Ontario and 518 (347 to 689) claims per 10,000 person-weeks in BC in the week of March 16, 2020 (S3 Table in S1 File). Similarly, ED visits decreased by 34 (30 to 38) visits per 10,000 person-weeks in Ontario and 17 (12 to 22) visits per 10,000 person-weeks in BC (S4 Table in S1 File). Sharp level declines in hospital admissions were also seen in both provinces [Ontario: -27% (-32% to -23%), $p<0.001$; BC: -21% (-26% to -16%), $p<0.001$] (Fig 1c). These corresponded to decreases of 4 (3 to 5) and 4 (3 to 4) admissions per 10,000 person-weeks in Ontario and BC, respectively (S5 Table in S1 File).

There was also a decline in visits to cancer clinics in Ontario [-11% (-16% to -6%), $p<0.001$], corresponding to a decrease of 2 (1 to 3) visits per 10,000 person-weeks (S1 Fig in S1 File). Of the resources analyzed with ITS, only home care services and visits to dialysis clinics did not demonstrate statistically significant decreases in March 2020 [home care $p = 0.11$ and $p = 0.70$ in Ontario and BC, respectively; dialysis $p = 0.58$, Ontario] (S2 Fig in S1 File).

### Weekly resource use (by category) in 2020 versus historical averages

Fig 2a and 2b show the overlaid patterns of select resource categories in each week of 2020 normalized to their two-year pre-pandemic average in Ontario and BC, highlighting the different ways the pandemic impacted each resource. In the first pandemic week, same day surgery visits sharply decreased by 69% in Ontario and 48% in BC, compared with the weekly averages of 2018 and 2019, while ED visits and hospital admissions decreased by 28% and 35% in Ontario, respectively, and 31% and 38% in BC. Even with gradual returns toward pre-pandemic levels, many resources notably remained below full capacity by December 2020. In both Ontario and BC, ED visits were still at least 10% (Ontario) and 7% (BC) below pre-pandemic levels from July 2020 to September 2020, only to decrease again for the remainder of 2020.

The notable exception was a transient increase of up to 48% in the number of prescription drug claims in May and June 2020 in Ontario, coinciding with a change in dispensing policies (limited to a 30-day supply rather than 100-day supply per prescription to limit drug

**Table 1. Characteristics of the individuals in the first week of 2020 in Ontario and British Columbia.**

| Characteristic | Ontario (N = 14,973,016) | British Columbia (N = 5,123,554) |
|---|---|---|
| **Age in years** | | |
| Mean (SD) | 41.2 (23.0) | 42.2 (23.0) |
| Median (IQR) | 41 (23–59) | 42 (24–60) |
| **Sex (N, (%))** | | |
| Female | 7,582,509 (50.6%) | 2,590,900 (50.6%) |
| Male | 7,390,507 (49.4%) | 2,532,555 (49.4%) |
| Unknown/missing | | 101 (0.0%) |
| **Rural/Urban residence[1] (N (%))** | | |
| Urban | 13,429,437 (89.7%) | 4,547,778 (88.8%) |
| Rural/small town | 1,506,012 (10.1%) | 542,364 (10.6%) |
| missing | 37,567 (0.3%) | 33,414 (0.65) |
| **Neighbourhood income quintile (N, (%))** | | |
| 1 (lowest) | 2,932,526 (19.6%) | 1,006,951 (19.6%) |
| 2 | 2,926,151 (19.5%) | 1,005,234 (19.6%) |
| 3 | 3,009,569 (20.1%) | 1,027,220 (20%) |
| 4 | 3,031,126 (20.2%) | 1,040,260 (20.3%) |
| 5 (highest) | 3,030,496 (20.2%) | 954,329 (18.6%) |
| Missing | 43,148 (0.3%) | 89,562 (1.2%) |
| **Ontario Marginalization Index[2] (N (%))** | | |
| Missing all dimensions | 591,376 (3.9%) | |
| **Deprivation** | | |
| 1 (lowest) | 3,336,479 (22.3%) | |
| 2 | 3,042,928 (20.3%) | |
| 3 | 2,735,771 (18.3%) | |
| 4 | 2,601,692 (17.4%) | |
| 5 (highest) | 2,664,770 (17.8%) | |
| **Dependency** | | |
| 1 (lowest) | 4,001,343 (26.7%) | |
| 2 | 2,928,380 (19.6%) | |
| 3 | 2,507,571 (16.7%) | |
| 4 | 2,402,262 (16.0%) | |
| 5 (highest) | 2,542,084 (17.0%) | |
| **Instability** | | |
| 1 (lowest) | 3,215,152 (21.5%) | |
| 2 | 2,746,005 (18.3%) | |
| 3 | 2,626,851 (17.5%) | |
| 4 | 2,571,331 (17.2%) | |
| 5 (highest) | 3,222,301 (21.5%) | |
| **Ethnic diversity** | | |
| 1 (lowest) | 2,216,225 (14.8%) | |
| 2 | 2,353,671 (15.7%) | |
| 3 | 2,563,036 (17.1%) | |
| 4 | 3,059,575 (20.4%) | |
| 5 (highest) | 4,194,864 (28.0%) | |
| **ADG score[3] (N (%))** | | |
| -40 to -20 | 34,937 (0.2%) | 14,724 (0.3%) |
| -19 to 0 | 5,815,952 (38.8%) | 1,897,905 (37%) |

*(Continued)*

**Table 1.** (Continued)

| Characteristic | Ontario (N = 14,973,016) | British Columbia (N = 5,123,554) |
|---|---|---|
| 1 to 20 | 7,535,924 (50.3%) | 2,554,776 (49.9%) |
| 21 to 40 | 1,008,343 (6.7%) | 368,381 (7.2%) |
| 41 to 60 | 116,868 (0.8%) | 28,753 (0.6%) |
| 61 to 80 | 8,248 (0.1%) | 904 (0.0%) |
| missing | 452,744 (3.0%) | 258,113 (5.0%) |
| **Ontario Local Health Integration Network (N (%))** | | |
| Erie St. Clair | 692,031 (4.6%) | |
| South West | 1,044,956 (7.0%) | |
| Waterloo Wellington | 852,754 (5.7%) | |
| Hamilton Niagara Haldimand Brant | 1,550,091 (10.4%) | |
| Central West | 1,058,630 (7.1%) | |
| Mississauga Halton | 1,319,817 (8.8%) | |
| Toronto Central | 1,355,793 (9.1%) | |
| Central | 2,041,614 (13.6%) | |
| Central East | 1,711,708 (11.4%) | |
| South East | 524,688 (3.5%) | |
| Champlain | 1,454,377 (9.7%) | |
| North Simcoe Muskoka | 522,892 (3.5%) | |
| North East | 591,516 (4.0%) | |
| North West | 252,144 (1.7%) | |
| Shared/Split | < = 5[4] | |
| missing | < = 5[4] | |
| **British Columbia Health authority (N (%))** | | |
| Interior | | 829,497 (16.2%) |
| Fraser | | 1,901,542 (37.1%) |
| Vancouver Coastal | | 1,213,904 (23.7%) |
| Vancouver Island | | 853,537 (16.7%) |
| Northern | | 296,183 (5.8%) |
| missing | | 28,893 (0.5%) |

Abbreviations: ADGs -Adjusted Diagnostic Groups; IQR- interquartile range; N- number of people; SD- standard deviation.

[1] Rural/urban residence was determined using the postal code indicator [58] in British Columbia and the Statistics Canada definition [59] in Ontario. Rural postal codes are for areas serviced by rural route drivers and/or postal outlets [58]. Statistics Canada defines rural and small town as areas with populations of less than 10,000 persons and outside the commuting zones of urban centres with populations of 10,000 or more, based on the current census population count [59].

[2] The Ontario Marginalization Index is an area-level measure that was developed using 42 variables from the Canada Census. It includes four dimensions, measured in quintiles: material deprivation (based on income, parental status, education, and employment), ethnic diversity (proportion of individuals self-identifying as a visible minority, and recent immigrants), dependency (proportion of individuals aged 0 to 14 and 65 years and older, or unemployed/unpaid), and residential instability (based on the type and density of dwellings, families, and individuals) [60].

[3]The Aggregated Diagnosis Group (ADG) score assigns positive or negative weights to the ADGs derived from John's Hopkins' Adjusted Clinical Groups® System (ACG®) version 10 Aggregated Diagnosis Groups (ADGs) [65], depending on the magnitude and direction of their association with the probability of one-year mortality [64]. Because calculation of the ADG score requires data covering the previous year, we were unable to report ADG scores for newborns and other people who had less than one year of OHIP or MSP coverage before the first week of 2020.

[4]Small cell sizes cannot be reported to comply with ICES privacy rules.

stockpiling). In BC, the number of prescription drug claims remained relatively stable, with only a small increase of 19% in the first week of the pandemic in 2020.

Three distinct patterns were observed: I) a sharp decline and slow recovery, mostly to lower than pre-pandemic levels (as above); II) a saw-tooth pattern in same-day surgery, illustrating

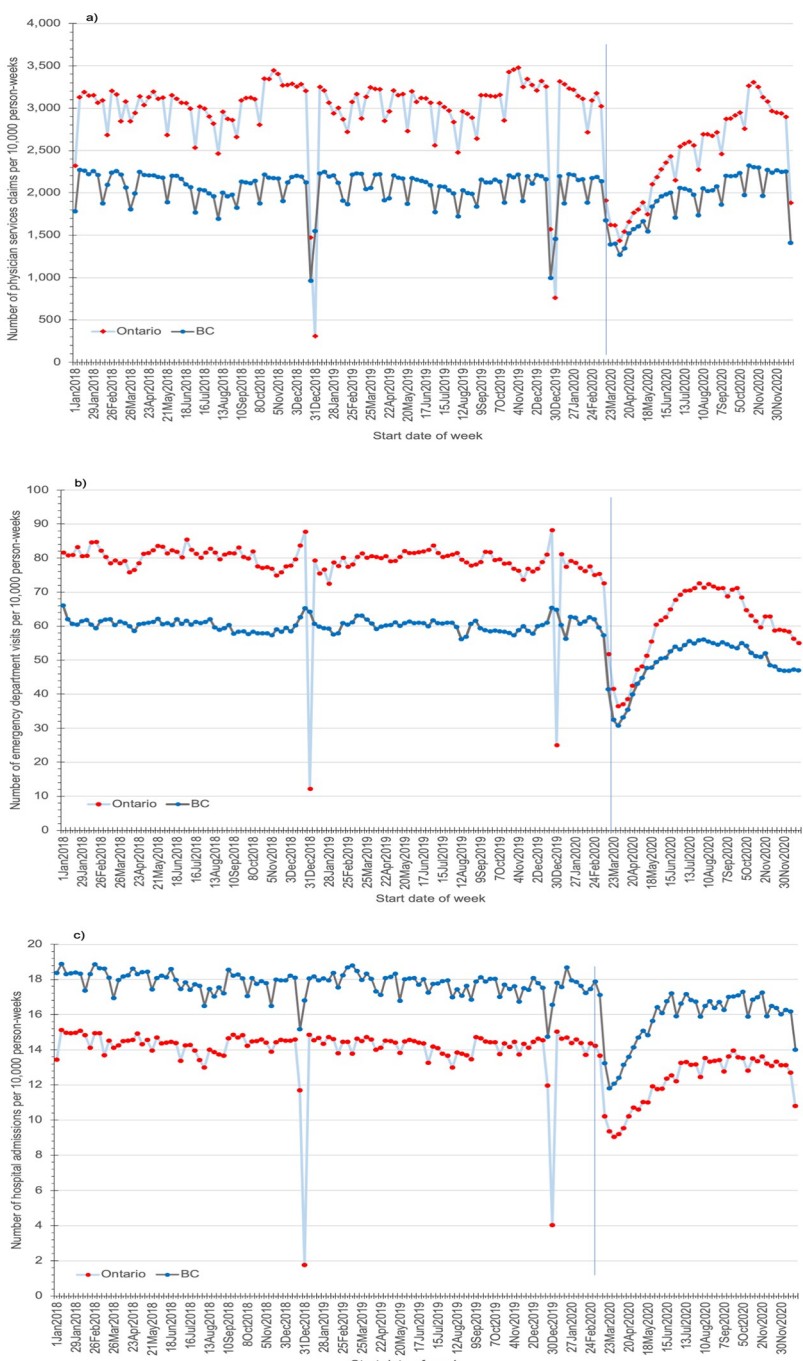

**Fig 1.** Number of a) physician services claims, b) emergency department visits, and c) hospital admissions per 10,000 person-weeks in each ISO week from January 1, 2018 to December 27, 2020 in Ontario and British Columbia (BC). These data were analyzed using interrupted time series (ITS). The vertical line in each graph indicates the first week of the pandemic, March 16 to March 22, 2020.

the stop-and-start nature of the ramp-down and ramp-up strategies as well as seasonal variability (e.g., statutory holidays); and III) stable resources that fluctuated by a relatively small margin, such as home care in both provinces, prescription drugs in BC, and dialysis and cancer clinics in Ontario.

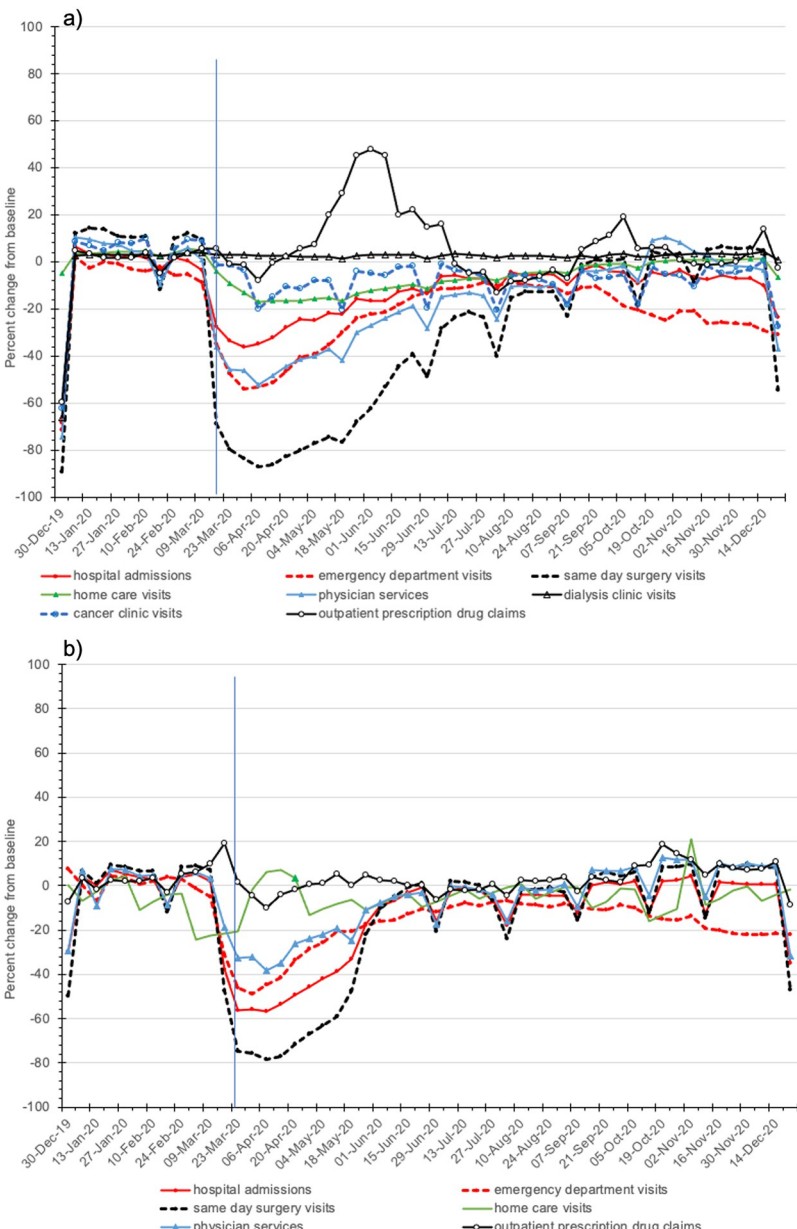

**Fig 2.** Percentage change in use per 10,000 person-weeks in 2020 compared with the average weekly use in 2018 and 2019 in (a) Ontario and (b) British Columbia for hospital admissions, same day surgery visits, laboratory tests (do not include SAR-CoV-2 tests), long-term care admissions, outpatient prescription drug claims, and cancer clinic and dialysis clinic visits (Ontario only). This shows the overlaid performance of select resource categories in each ISO week of 2020 normalized to their two-year pre-pandemic average in Ontario and British Columbia. The vertical line indicates the first week of the pandemic, March 16 to March 22, 2020.

## Healthcare costs

The ITS analyses showed that in the week of March 16 to 23, 2020, total direct healthcare costs declined by 27% (23% to 31%, p<0.001) in Ontario and 18% (14% to 22%, p<0.001) in BC, compared to the final pre-pandemic week. These equate to decreases of $161,407 ($140,009 to $182,805) and $72,689 ($49,126 to $96,253) per 10,000 person-weeks in each province, respectively (Fig 3, S6 Table in S1 File).

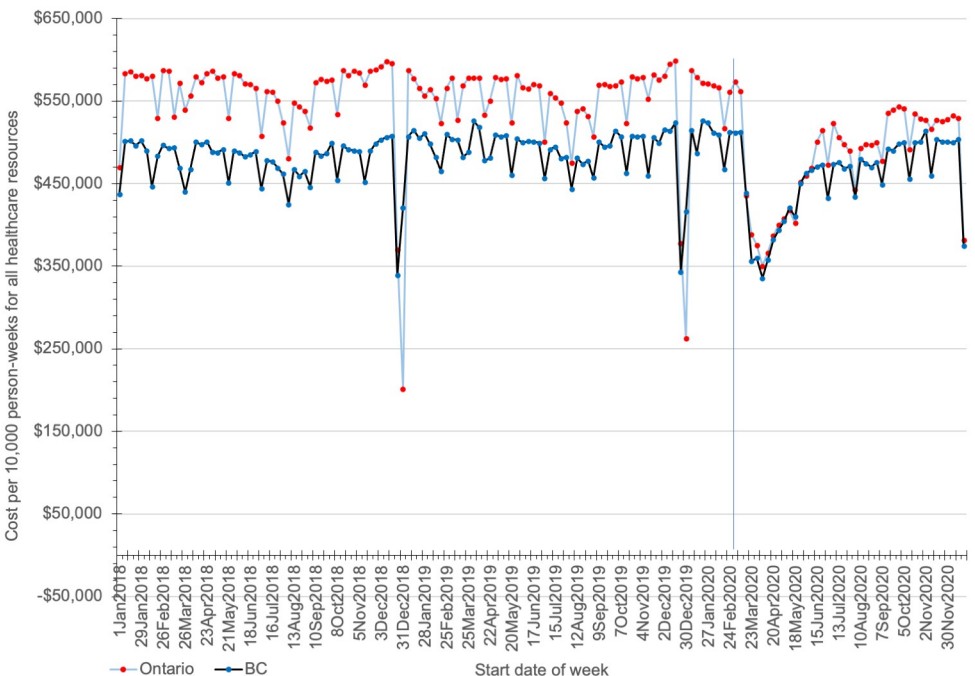

**Fig 3. Total healthcare costs per 10,000 person-weeks (in 2020 CAD) in each ISO week from January 1, 2018 to December 27, 2020 in Ontario and British Columbia (BC).** These data were analyzed using interrupted time series (ITS). The vertical line indicates the first week of the pandemic, March 16 to March 22, 2020.

As shown in Tables 2 and 3, the total costs for the healthcare services included in this study in the first quarter of 2020 were $10.1 billion in Ontario and $3.2 billion in BC, 6.2% and 2.5% lower than the average quarterly cost in 2018 and 2019, respectively (S7 and S8 Tables in S1 File). The total costs in the second quarter were approximately $8.3 billion in Ontario and $2.8 billion in BC, 23% and 12% lower than pre-pandemic, respectively. In Ontario, total costs increased to $9.8 billion and almost $10.2 billion in the third and fourth quarters, respectively, but the latter was still 5.5% below the average pre-pandemic quarterly cost. In BC, total costs increased in the third and fourth quarters of 2020 to $3.1 billion and almost $3.3 billion respectively, approximately equal to and almost 4% higher than pre-pandemic. Some of the resources for which quarterly costs decreased during the pandemic appeared to return to pre-pandemic levels more than others. In Ontario, costs for same day surgery and physician services (primary care providers) were as high as pre-pandemic levels by the last quarter of 2020. In BC, the costs for same day surgery and services of both specialist and primary care providers were higher than in the pre-pandemic years in the last two quarters of 2020.

In both provinces, hospitalization, physician services, and outpatient prescription drugs made up the largest components of total cost, and hospitalization was the main driver for the absolute cost declines in the second quarter of 2020 (Tables 2 and 3). In Ontario, the quarterly costs for visits to cancer and dialysis clinics, outpatient prescription drugs and cancer drugs covered by the New Drug Funding Program [75] were higher in 2020 than in the previous two years.

## Direct healthcare costs for COVID-19

The total costs in 2020 for all healthcare services included in this study were approximately $12.4 billion in BC and $38.4 billion in Ontario. During this time, costs for cases with a

**Table 2. Total and resource-specific absolute costs (in 2020 CAD) per quarter in 2020 and percentage change from 2018 and 2019 for all individuals in Ontario.**
For readability, costs are rounded to the nearest thousand dollars; the percentages were calculated using costs estimated to the nearest dollar.

| Healthcare Resource | Time period[1] | | | |
|---|---|---|---|---|
| | 2020 Q1 (% change from baseline[2]) | 2020 Q2 (% change from baseline[2]) | 2020 Q3 (% change from baseline[2]) | 2020 Q4 (% change from baseline[2]) |
| Hospitalization | 2,292,816,000 (-1.30) | 1,866,980,000 (-19.63) | 2,196,786,000 (-5.44) | 2,096,122,000 (-9.77) |
| Emergency Department visits | 393,967,000 (-11.54) | 318,317,000 (-28.52) | 409,792,000 (-7.98) | 374,548,000 (-15.90) |
| Same Day Surgery | 374,287,000 (-6.22) | 147,305,000 (-63.09) | 351,769,000 (-11.86) | 419,817,000 (5.19) |
| Physician services -PCPs | 567,953,000 (-6.34) | 524,437,000 (-13.52) | 611,157,000 (0.78) | 664,930,000 (9.65) |
| Physician services—specialists | 1,618,475,000 (-10.45) | 1,169,682,000 (-35.28) | 1,643,082,000 (-9.08) | 1,800,056,000 (-0.40) |
| Non-physicians services[3] | 33,994,000 (-23.73) | 8,742,000 (-80.38) | 37,154,000 (-16.62) | 40,527,000 (-9.05) |
| Cancer clinic visits | 399,238,000 (15.50) | 365,542,000 (5.75) | 377,951,000 (9.34) | 383,799,000 (11.03) |
| Dialysis clinic visits | 186,204,000 (-1.86) | 204,059,000 (7.55) | 204,555,000 (7.81) | 206,629,000 (8.90) |
| Outpatient clinic visits[4] | 604,722,000 (-22.02) | 222,653,000 (-71.29) | 392,279,000 (-49.41) | 458,938,000 (-40.82) |
| Laboratory tests[5] | 147,212,000 (-31.23) | 83,230,000 (-61.12) | 141,209,000 (-34.04) | 149,342,000 (-30.24) |
| Outpatient prescription drug claims[6] | 1,630,616,000 (0.37) | 1,664,768,000 (2.48) | 1,657,124,000 (2.01) | 1,789,378,000 (10.15) |
| New Drug Funding Program drugs[7] | 154,663,000 (22.33) | 156,457,000 (23.75) | 165,206,000 (30.67) | 165,894,000 (31.22) |
| Home Care services | 898,032,000 (0.63) | 820,858,000 (-8.02) | 857,717,000 (-3.89) | 899,304,000 (0.77) |
| Long-term Care | 446,526,000 (-26.19) | 374,418,000 (-38.11) | 313,848,000 (-48.12) | 256,946,000 (-57.53) |
| Complex Continuing Care | 202,774,000 (-3.42) | 187,796,000 (-10.56) | 185,961,000 (-11.43) | 181,155,000 (-13.72) |
| Mental health inpatient care | 159,682,000 (-5.92) | 126,533,000 (-25.45) | 140,870,000 (-17.00) | 137,599,000 (-18.93) |
| SARS-CoV-2 tests | 2,900,000 (n/a) | 58,902,000 (n/a) | 109,424,000 (n/a) | 161,485,9000 (n/a) |
| TOTAL | 10,114,061,000 (-6.17) | 8,300, 682,000 (-22.99) | 9,795,884,000 (-9.12) | 10,186,469,000 (-5.50) |

[1] Time periods comprise 13 weeks

[2] Baseline equals the average cost per quarter in 2018 and 2019, based on S8 Table in S1 File.

[3] Non-physician services covered by the Ontario Health Insurance Plan include care provided by optometrists, podiatrists, physiotherapists, and other allied health professionals, for eligible individuals [74]

[4] Outpatient hospital clinics, other than dialysis and cancer

[5] Excludes SARS-CoV-2 tests

[6] Outpatient prescription drugs for eligible individuals, aged 65 years and older or who meet other criteria [66]

[7] The New Drug Funding Program reimburses cancer centres for expensive intravenous cancer chemotherapy [75, 76]

Abbreviations: PCPs: Primary care providers; NDFP- New Drug Funding Program; Q—Quarter

positive SARS-Cov-2 test from diagnosis to death or censoring were approximately $89.8 million in BC and $504 million in Ontario, representing 0.7% and 1.3% of total costs in BC and Ontario, respectively (S9 Table in S1 File).

## Discussion

Our study systematically examined the initial effects of the COVID-19 pandemic on total direct healthcare resource use and costs in two Canadian provinces with universal healthcare, demonstrating the heterogeneous impact of the pandemic across healthcare resources. Early in the pandemic, healthcare resources were reallocated to meet the anticipated and real demands of patients with COVID-19 [5, 6, 8]. However, while we observed immediate decreases in use of physician services, hospital admissions, and emergency department visits, it is reassuring that visits to dialysis clinics, an essential and life-sustaining resource, did not appear to decrease during the first year of the COVID-19 pandemic. Similarly, the rate of home care

**Table 3. Total and resource-specific absolute costs per quarter in 2020 (in 2020 CAD) and percentage change from 2018 and 2019 for all individuals in British Columbia.** For readability, costs are rounded to the nearest thousand dollars; the percentages were calculated using costs estimated to the dollar.

| Healthcare Resource | Time period[1] | | | |
|---|---|---|---|---|
| | 2020 Q1 (% change from baseline[2]) | 2020 Q2 (% change from baseline[2]) | 2020 Q3 (% change from baseline[2]) | 2020 Q4 (% change from baseline[2]) |
| Hospitalization | 1,035,153,000 (3.61%) | 856,947,000 (-14.22%) | 975,646,000 (-2.34%) | 993,398,000 (-0.57%) |
| Emergency Department visits | 117,862,000 (-2.54%) | 92,464,000 (-23.54%) | 113,586,000 (-6.08%) | 103,295,000 (-14.59%) |
| Same day surgery visits | 168,190,000 (-1.24%) | 112,204,000 (-34.12%) | 179,589,000 (5.45%) | 192,480,000 (13.02%) |
| Physician services -PCPs | 274,998,000 (1.68%) | 241,926,000 (-10.55%) | 279,829,000 (3.46%) | 286,545,000 (5.95%) |
| Physician services -Specialists | 204,454,000 (1.13%) | 202,343,000 (0.09%) | 209,495,000 (3.62%) | 230,057,000 (13.79%) |
| Other Medical Services Plan services[3] | 439,771,000 (1.26%) | 318,661,000 (-26.63%) | 408,462,000 (-5.95%) | 433,980,000 (-0.07%) |
| Medical Services Plan (all) | 919,223,000 (1.35%) | 762,930,000 (-15.88%) | 897,785,000 (-1.01%) | 950,582,000 (4.81%) |
| Outpatient prescription drug claims | 839,383,000 (5.72%) | 803,697,000 (1.23%) | 824,063,000 (3.79%) | 865,868,000 (9.06%) |
| Home Care services | 140,628,000 (-6.41%) | 148,747,000 (-1.01%) | 148,850,000 (-0.94%) | 149,161,000 (-0.73%) |
| SARS-CoV-2 tests | 44,000 (n/a) | 72,000 (n/a) | 251,000 (n/a) | 1,780,000 (n/a) |
| TOTAL | 3,220,483,000 (2.52%) | 2,777,062,000 (-11.60%) | 3,139,771,000 (-0.05%) | 3,256,564,000 (3.66%) |

[1] Time periods comprise 13 weeks

[2] Baseline equals the average cost per quarter in 2018 and 2019, based on S9 Table in S1 File.

[3] Other Medical Services Plan services include records for diagnostic procedures, procedures and other services

Abbreviations: PCPs: Primary care providers; Q- quarter

services did not significantly change, and the decline in visits to cancer clinics in Ontario was smaller than the declines of other resources, suggesting some degree of prioritization of home care and cancer care, although we did not examine the types of services provided.

Several other studies examined changes in cancer care during the COVID-19 pandemic. A study in Ontario found that visits for cancer systemic therapy and radiation therapy were 3.5% and 21% lower, respectively, in the year after March 31, 2020 than in the prior year [77]. Another Canadian study predicted increased cancer-related deaths in the long-term as a result of diagnostic and treatment delays [78]. In England, the mean number of weekly radiation therapy courses delivered by the National Health Service decreased by 20%, 6% and 12% in April, May and June 2020, respectively, compared to the same months in 2019 [22]. The greatest decreases were in treatment for prostate and non-melanoma skin cancers. In both Ontario and England, increased use of hypofractionated radiation regimens explained some of the decreased number of visits [22, 77]. With respect to cancer detection, studies in Japan and Brazil reported reductions in the numbers of mammograms performed and increases in the proportions of patients diagnosed with advanced breast cancer in the pandemic period compared to before [16, 17]. Therefore, there is some evidence from these studies and ours that efforts to maintain cancer treatment were made early in the pandemic, while cancer screening was reduced. The long-term impacts of cancer care delays and deferrals leading to more severe presentations are yet to be revealed [16, 78]. Similar to other studies [26, 27], we found that the rates of ED visits significantly declined at the start of the pandemic compared to the last prepandemic week, by 41% in Ontario and 24% in BC, and never returned to pre-pandemic levels in 2020. This is consistent with ED avoidance by people with milder conditions, diversion of human and diagnostic resources away from urgent care, as well as significant decreases in infections associated with lockdown measures. With respect to changes in patterns of emergency medical care, a study in the Canadian city of Calgary, Alberta, reported 35% and 59%

fewer patients presenting to adult and pediatric hospital EDs, respectively from December 2019 to June 2020 compared with the same months in the previous two years [26]. Corresponding reductions of 26% to 30% in overall consultations were observed in 20 emergency departments in Germany during the first 3 waves of COVID-19 from March 2020 to June 2021 [27].

Similarly, analyses from 18 hospitals in BC reported decreases of 57% and 70% in visits to pediatric and general hospital EDs, respectively, by children aged 0 to 18 years during the first 6 weeks of the pandemic compared with the same weeks in the previous year [10]. One study in Ontario examined ED and hospital admissions with mental health diagnoses from January 2019 to March 2021, and found sharp decreases of 37% and 30% in rates of ED visits and hospital admissions, respectively, in April 2020 compared with April 2019 [79]. Rates of ED visits for mental health diagnoses increased by July 2020 only to gradually decrease again.

In aggregate, these findings demonstrate that the healthcare system was able to continue providing life-sustaining care to patients who needed it, for example those receiving hemodialysis. However, the use of services appeared to decline according to the severity of conditions they are intended to treat, with greater reductions in the use of elective procedures, and preventive, screening, and routine care than urgent and semi-urgent services [35, 80, 81]. Furthermore, after sharp initial declines in use, recovery was slow for many healthcare resources, and their use did not return to pre-pandemic levels by December 2020. Notably, the recovery was much less than needed to manage the backlogs (areas above the curves with negative values, and below the zero-axis), which raises concerns about the residual unmet healthcare needs of the population. This could be a compounding effect if unmet needs resulted in acute events that could have been avoided with earlier care.

The direct healthcare costs for COVID-19 cases in 2020 corresponded to a modest share of the healthcare costs for the full populations in Ontario and BC, at 1.3% and 0.7%, respectively. The magnitude of the resources used by COVID-19 patients was far outweighed by reduced service utilization in the general population. However, the economic impact of COVID-19 includes costs for vaccines, personal protective equipment, improving indoor air quality, paid sick leave, and other support beyond the direct healthcare costs measured in our study [82, 83]. Furthermore, much of the human burden of COVID-19 and long COVID-19 have not yet been estimated [84], and may be beyond our metrics.

In anticipation of future health crises, it is critical to ensure an equitable allocation of resources and a "reserve" within the healthcare system. A resilient healthcare system must respond to the needs of COVID-19 patients while preserving standards of care for all patients, and preserving access to care for its community [85]. Resilient hospitals and healthcare systems must implement changes that will be able to meet any challenges they may encounter and that will be useful even during normal times [85]. Our data suggested that such a reserve was not present within the Canadian universal healthcare system. In planning for COVID-19 recovery phases, integration and communication among policy makers, health economists, and clinicians across services may be necessary to help the healthcare system adapt to crises [86], inform resource allocation, and prioritize recovery efforts.

### Strengths and limitations

Strengths of this study include the use of comprehensive population-wide data which included almost all healthcare resource use in two populous Canadian provinces with provincial publicly-funded healthcare. This study used standardized methods to estimate resource use and costs [87, 88], ensuring valid outcome measures. However, some of the absolute differences in the service rates between provinces may be due to differences in the scope of the data available in Ontario and BC and underlying program structure.

Despite the comprehensiveness of our patient-level data, we were not able to measure the unmet healthcare needs of those who did not access resources. Furthermore, we could not capture the expenditures related to health systems, such as costs associated with infrastructure changes or the rapid adoption of telehealth initiatives. While health system directives were a major factor in the observed changes, some of the reductions in healthcare use may also have been driven by patients' avoidance of medical settings [80]. However, social distancing behaviours, directives, or patients' motivation are likely to have acted as mediators, rather than confounders, of the changes in healthcare costs and use that we observed. That is, the behavioural changes and policies are part of the COVID-19 pandemic effect, rather than confounding the true effect.

Although the ITS analysis allowed us to project a counterfactual (based on pre-pandemic secular trends had the pandemic not occurred), we cannot comment on the appropriateness of this benchmark. However, this is likely to represent a conservative bias as a return to pre-COVID-19 trends and levels does not account for the backlog accrued during COVID-19. Our study covered the immediate responses of the healthcare system to the shocks of the first two waves of COVID-19. Subsequent time periods with waves of new COVID-19 variants and relaxation of public health measures would require different methods to answer different questions. Finally, even though the experiences in Ontario and BC may not represent those in all of Canada or other countries, their similar patterns suggest that other provinces might have responded in much the same way to the initial challenges of the COVID-19 pandemic, and may serve as a blueprint for other publicly-funded healthcare systems in the world.

In summary, our study confirms that the overall use of physician services, hospital admissions, and emergency department visits greatly decreased in response to the initial COVID-19 pandemic, while life-sustaining dialysis care was maintained, as were home care services and cancer care in Ontario. Thus, resource optimization appeared to be appropriate to the degree of necessary care. However, recovery for many healthcare resources was slow after sharp declines, and the resultant backlogs raise concerns about residual unmet needs. Finally, in Ontario and BC, the direct healthcare costs for COVID-19 cases in 2020 represented were outweighed by reduced costs in the general population consequent to decreased resource use.

## Conclusions

We observed decreased costs and rates of use for most healthcare resources in two Canadian provinces with predominantly publicly-funded healthcare following the declaration of the COVID-19 pandemic and the associated initial mitigation measures. These changes may have compromised the care of many non-COVID-19 patients in efforts to prepare for the anticipated demand of care for patients with COVID-19. These findings highlight the need for more flexible and sustainable systems that can ensure timely access to adequate care for all individuals. Monitoring to detect and immediately address backlogs could avoid the need for supra-normal levels of activity to "catch up" the system. Further research on the potential consequences of delayed and cancelled medical care will be necessary to mitigate risks in access and quality of healthcare services, address backlog recovery, and avoid healthcare system collapse.

## Supporting information

**S1 File.**
(DOCX)

## Acknowledgments

We acknowledge the contributions of Dr. Murray Dale Krahn, who passed away on July 1, 2022. Soon after COVID-19 was declared a pandemic, Dr. Krahn foresaw the potential impact that it might have on our healthcare system, and realized the importance of examining system-wide responses in these extraordinary times. He asked not only how the health system would care for patients with COVID-19, but also what would happen to other patients, when health-care pivoted to prioritize COVID-19. This study is one product of his inspiration, guidance, and pioneering forays into health services research over the years, and we dedicate this paper to his honour.

This document used data adapted from the Statistics Canada Postal Code[OM] Conversion File Plus Version 7B, which is based on data licensed from Canada Post Corporation, and/or data adapted from the Ontario Ministry of Health Postal Code Conversion File, which contains data copied under license from ©Canada Post Corporation and Statistics Canada. Parts of this material are based on data and information compiled and provided by Ontario Ministry of Health (MOH). The analyses, conclusions, opinions and statements expressed herein are solely those of the authors and do not reflect those of the funding or data sources; no endorsement is intended or should be inferred. Parts of this material are based on data and information provided by Ontario Health (OH). The opinions, results, view, and conclusions reported in this paper are those of the authors and do not necessarily reflect those of OH. No endorsement by OH is intended or should be inferred. Parts of this material are based on data and/or information compiled and provided by CIHI and the Ontario Ministry of Health. The analyses, conclusions, opinions and statements expressed herein are those of the authors and do not reflect those of the funding or data sources; no endorsement is intended or should be inferred. Parts of this material are based on data adapted from Statistics Canada, the Canada Census, 2016. This does not constitute an endorsement by Statistics Canada of this product. We thank IQVIA Solutions Canada Inc. for use of their Drug Information File. We thank the Toronto Community Health Program Profiles Partnership for providing access to the Ontario Marginalization Index.

## Author Contributions

**Conceptualization:** Douglas C. Cheung, Karen E. Bremner, Stuart Peacock, Andrew B. Mendlowitz, Jennifer D. Walker, Murray D. Krahn, Girish S. Kulkarni.

**Formal analysis:** Seraphine Zeitouny, Reka E. Pataky, Priscila Pequeno, John Matelski, George Tomlinson.

**Funding acquisition:** Douglas C. Cheung, Karen E. Bremner, Stuart Peacock, Andrew B. Mendlowitz, Murray D. Krahn, Girish S. Kulkarni.

**Methodology:** Seraphine Zeitouny, Douglas C. Cheung, Reka E. Pataky, Priscila Pequeno, John Matelski, M. Elisabeth Del Giudice, Lauren Lapointe-Shaw, George Tomlinson, Carol Mulder, Teresa C. O. Tsui, Nathan Perlis, Murray D. Krahn, Girish S. Kulkarni.

**Project administration:** Karen E. Bremner.

**Supervision:** Stuart Peacock, Murray D. Krahn, Girish S. Kulkarni.

**Writing – original draft:** Seraphine Zeitouny, Douglas C. Cheung, Karen E. Bremner, Reka E. Pataky, Priscila Pequeno, John Matelski.

**Writing – review & editing:** Seraphine Zeitouny, Douglas C. Cheung, Karen E. Bremner, Reka E. Pataky, Priscila Pequeno, John Matelski, Stuart Peacock, M. Elisabeth Del Giudice,

Lauren Lapointe-Shaw, George Tomlinson, Andrew B. Mendlowitz, Carol Mulder, Teresa C. O. Tsui, Nathan Perlis, Jennifer D. Walker, Beate Sander, William W. L. Wong, Girish S. Kulkarni.

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
