## [Decision Letter · Decision Letter 0]

10 May 2023

PONE-D-23-03072The impact of the early COVID-19 pandemic on health care system resource use and costs in two provinces in Canada: An interrupted time series analysisPLOS ONE

Dear Dr. Cheung,

Thank you for submitting your manuscript to PLOS ONE. After careful consideration, we feel that it has merit but does not fully meet PLOS ONE’s publication criteria as it currently stands. Therefore, we invite you to submit a revised version of the manuscript that addresses the points raised during the review process.

We look forward to receiving your revised manuscript.

Kind regards,

Masoud Behzadifar

Academic Editor

PLOS ONE

Journal Requirements:

2. Please ensure that you have specified (1) whether consent was informed and (2) what type you obtained (for instance, written or verbal, and if verbal, how it was documented and witnessed). If your study included minors, state whether you obtained consent from parents or guardians. If the need for consent was waived by the ethics committee, please include this information.

“This study was supported by ICES, which is funded by an annual grant from the Ontario Ministry of Health (MOH) and the Ministry of Long-Term Care (MLTC). This document also used data adapted from the Statistics Canada Postal Code^OM ^Conversion File Plus Version 7B, which is based on data licensed from Canada Post Corporation, and/or data adapted from the Ontario Ministry of Health Postal Code Conversion File, which contains data copied under license from ÓCanada Post Corporation and Statistics Canada”

“This study was funded by a Canadian Institutes of Health Research (CIHR) operating grant: COVID-19 Rapid Research Funding Opportunity (funding reference number VR4 172774) to Dr. Krahn and Dr. Kulkarni.

This work was also undertaken, in part, thanks to funding from the Canada Research Chairs program to Dr. Krahn, Dr. Walker and Dr. Sander and an Ontario Early Researcher Award to Dr. Wong.

The Canadian Centre for Applied Research in Cancer Control is funded by the Canadian Cancer Society.

Role of the Funder/Sponsor

The analyses, conclusions, opinions and statements expressed herein are solely those of the authors and do not necessarily reflect those of the funding sources or data sources; no

endorsement is intended or should be inferred.”

Reviewers' comments:

Reviewer's Responses to Questions

**Comments to the Author**

1. Is the manuscript technically sound, and do the data support the conclusions?

Reviewer #1: Partly

Reviewer #2: No

2. Has the statistical analysis been performed appropriately and rigorously? 

Reviewer #1: N/A

Reviewer #2: Yes

3. Have the authors made all data underlying the findings in their manuscript fully available?

Reviewer #1: No

Reviewer #2: Yes

4. Is the manuscript presented in an intelligible fashion and written in standard English?

Reviewer #1: No

Reviewer #2: Yes

5. Review Comments to the Author

Reviewer #1: Dear Co-editor

I am pleased to share my comment for title: The impact of the early COVID-19 pandemic on health care system resource use and costs in two provinces in Canada: An interrupted time series analysis.

The study can be made more attractive and rich by making some of the mentioned corrections.

*It seems that the importance and necessity of the study has not been described correctly. There are many studies regarding the use of health system resources and its costs during epidemics.

*At the end of the introduction section, mention what innovation and new knowledge the current study provides for better understanding to health policymakers regarding child development.

*In the study method section: explain how the data collection. Describe in detail (DATASETS).

*Has the data analysis been done under the supervision of a statistician?

*In table one, the method of classifying the variables into several groups is explained.

*It seems that the way of presenting the results of the study is not coherent

*It seems that in the discussion section of the study, a small number of studies have been used for comparison, and also the results of the studies have been used for confirmation. First, a comparison should be made with the results of other studies, and then the reasons for conformity and non-conformity of the results with other studies should be mentioned.

*Regarding the reduction in the number of doctors' visits and the admission of patients in the hospital during the period of Covid19, as well as the lack of reduction in the number of patients in the dialysis department, sufficient explanations should be added.

*The conclusion of the study is one of the main parts. The final conclusion based on the findings of the study should be mentioned - in this section, explain what practical benefits the findings of the present study have for the health system.

*Acknowledgements: a paragraph is enough, it is too long and redundant.

*After studying the references: it can be seen that the majority of the references are related to Canadian studies and other studies in other countries have not been used.

*Very important question:

Did the level of communication between doctors and patients, as well as the level of communication between other health personnel and patients, not change during the period of the Covid19 epidemic? (Keeping distance of health personnel from patients, fewer patient visits, fear of covid19 and conflict with this disease) these confounding variables did not have an effect on the amount of use of health services as well as costs???

Reviewer #2: The manuscript was well presented from title to recommendations, it has clear lines to flow. They have separared the costs' decline due to the mitigation measures and again the covid related costs, which directly answers the aim of the research. Generally fine with statistical methods used, as the data used was available, starting from descriptive to autocorrelations was typical measure to use in such data, but due to the huge magnitude of the data some errors can be detected (which I did't focus that much).

6. PLOS authors have the option to publish the peer review history of their article (what does this mean?). If published, this will include your full peer review and any attached files.

Reviewer #1: No

Reviewer #2: No

---

## [Author Response · Author response to Decision Letter 0]

23 Jun 2023

PONE-D-23-03072

The impact of the early COVID-19 pandemic on healthcare system resource use and costs in two provinces in Canada: An interrupted time series analysis

RESPONSES TO THE REVIEWERS 

1. Is the manuscript technically sound, and do the data support the conclusions?

Reviewer #1: Partly

Reviewer #2: No

REPLY: We hope that our revisions have improved the manuscript. In particular, we have reviewed our Results, Discussion, and Conclusions to ensure that they are supported by the data. Please see our responses to the reviewers’ specific queries below.

2. Has the statistical analysis been performed appropriately and rigorously? 

Reviewer #1: N/A

Reviewer #2: Yes

REPLY: Thank you for the replies. The study team included two biostatisticians and three data analysts with statistical expertise. They were responsible for the statistical analyses. Please see our response to Reviewer 1, comment 4.

3. Have the authors made all data underlying the findings in their manuscript fully available?

Reviewer #1: No

Reviewer #2: Yes

REPLY: Please see the “Data Availability Statements” at the end of the manuscript, on page 30, and copied below. Because the data include personal health information, there are some established restrictions to the open sharing of the data.

“In Ontario, the dataset from this study is held securely in coded form at ICES. While legal data sharing agreements between ICES and data providers (e.g., healthcare organizations and government) prohibit ICES from making the dataset publicly available, access may be granted to those who meet pre-specified criteria for confidential access, available at www.ices.on.ca/DAS. The full dataset creation plan and underlying analytic code are available from the authors upon request, understanding that the computer program may rely on coding templates or macros that are unique to ICES and therefore either inaccessible or may require modification.

In British Columbia, access to data provided by the Data Steward(s) is subject to approval, but can be requested for research projects through the Data Steward(s) or their designated service providers.”

4. Is the manuscript presented in an intelligible fashion and written in standard English?

Reviewer #1: No

Reviewer #2: Yes

REPLY: We hope that, in addressing the reviewers’ comments, the revised manuscript is presented in an intelligible fashion. We have clarified the aims of the study in response to Reviewer 1, comment 1. We have added information about the data and their collection in response to Reviewer 1, comment 3, and revised the footnote to Table 1 in response to Reviewer 1, comment 5. The Discussion has been revised in response to Reviewer 1, comment 7. The scope of the reference list is now more international, as suggested by Reviewer 1, comment 11. 

Comments from Reviewer #1, and replies: 

1. It seems that the importance and necessity of the study has not been described correctly. There are many studies regarding the use of health system resources and its costs during epidemics.

REPLY: The reviewer is correct that many studies have examined the use and costs of specific healthcare resources during COVID-19, and we acknowledge this in the Introduction, lines 116 to 117. In the revised manuscript, we have added additional recent references regarding the use of healthcare resources during COVID-19, including references from several countries other than Canada (to address comment 11, below). 

Critically, however, we did not find another study which included all healthcare resources in a population-based sample. The advantages of this population-based study are in examining the changes in healthcare resource use and costs in two provinces with publicly-funded healthcare systems. Moreover, we leveraged the availability and comprehensiveness of the administrative healthcare databases in these provinces that enabled this research. Our study provides valuable information about the totality of the healthcare system response by taking this (broader) scope of healthcare services in the context of each other. Evaluating change in one area of resource use as compared and contrasted with other domains allows for a holistic understanding of how the healthcare system reacted and what resources seem to have been prioritized within the pandemic response. This interaction between healthcare resources addresses a substantial and important information gap versus prior studies which viewed these resources in isolation, or in the context of a single disease or organ system. We have revised the aim of our study to clarify and emphasize this important distinction, in the last paragraph of the Introduction, as follows:

“Thus, the aim of our study was to assess the initial impact of COVID-19 directives and related mitigation measures on total healthcare costs and resource use across inpatient, outpatient, and long-term care in a concerted fashion for the populations in two provinces within the Canadian universal health system. This broad scope of healthcare services addresses an important information gap by allowing comparisons across resources for a holistic understanding of how the healthcare system reacted and which resources were prioritized within the pandemic response. Specifically, we assessed the changes in the trends of publicly-funded healthcare services use and direct healthcare costs in Ontario and BC.”

2. At the end of the introduction section, mention what innovation and new knowledge the current study provides for better understanding to health policymakers regarding child development.

REPLY: Thank you for this suggestion. However, please note that our study did not focus on pediatric care or child development, or identify healthcare services or costs for children. Therefore, we are unable to suggest any innovation or knowledge regarding child development.

3. In the study method section: explain how the data collection. Describe in detail (DATASETS).

REPLY: Thank you for this suggestion. The healthcare administrative datasets used in this study are comprised of data routinely collected by the provincial Ministries of Health in the administration of the provincial universal, publicly-funded health care systems, and submitted to ICES (Ontario) and Population Data BC (British Columbia). We have added this to the Methods section, in the Data sources and settings subsection, lines 129 to 131 in the clean revised manuscript, and cited references as appropriate, as follows:

“In both provinces, data are collected for administrative purposes by the provincial Ministries of Health, and under the provincial universal health insurance plans [37, 38].’

Please note that the databases used in the study are described in S1 Table in the supplementary material, with references.

The Ontario administrative data from ICES and the data from British Columbia held at Population Data BC, are routinely used in numerous studies on healthcare. For examples, please see the following: https://www.ices.on.ca/Publications/Journal-Articles and https://www.popdata.bc.ca/ria

4. Has the data analysis been done under the supervision of a statistician?

REPLY: Thank you for asking about statistical expertise. The data analyses were done in consultation with two biostatisticians who are co-authors on the paper (GT, JM). The time series analyses were conducted by a biostatistician (JM). Other data analyses were performed by data analysts with statistical experience (PP in Ontario, and SZ and REP in BC). The final paper was approved by all co-authors, including the statisticians and data analysts. 

5. In table one, the method of classifying the variables into several groups is explained.

REPLY: Thank you for this comment. The reviewer is correct that the variables shown in Table 1 are described in the first paragraph of the “Study Outcomes and Statistical Analysis” section. More details to explain the variables are in the footnotes below Table 1. In reviewing these, we noticed that rural and urban were not well-defined. We have added these definitions, with a new reference, in the revised manuscript in the footnotes below Table 1, in lines 234 to 238 in the clean revised manuscript. The first footnote below Table 1 now reads: 

“Rural/urban residence was determined using the postal code indicator [58] in British Columbia and the Statistics Canada definition [59] in Ontario. Rural postal codes are for areas serviced by rural route drivers and/or postal outlets [58]. Statistics Canada defines rural and small town as areas with populations of less than 10,000 persons and outside the commuting zones of urban centres with populations of 10,000 or more, based on the current census population count [59]. 

6. It seems that the way of presenting the results of the study is not coherent.

REPLY: Thank you for the comment. To improve the readability of the manuscript, we have carefully reviewed the presentation of the Results to ensure that the results of each analysis are logical and follow from the Methods. We have revised the headings of some sections of the Results for clarity. There are several tables and figures in the manuscript and in the Supplemental Material. 

Please note that we have followed PLOS One’s formatting specifications and each figure caption appears directly after the paragraph in which it is first cited in the manuscript. The figure titles are in bold type. The figure will be placed with the figure caption in the published paper. This will greatly improve the presentation of the results.

7. It seems that in the discussion section of the study, a small number of studies have been used for comparison, and also the results of the studies have been used for confirmation. First, a comparison should be made with the results of other studies, and then the reasons for conformity and non-conformity of the results with other studies should be mentioned.

REPLY: We have revised the Discussion as suggested by the Reviewer. The first paragraph of the Discussion summarizes our results, including potential explanations (please see our reply to the next comment). The next three paragraphs report the results of other Canadian and international studies that examined cancer care, emergency department visits, and hospital admissions. Then we circle back to our results and suggest explanations for the results of our studies and those of the other studies.

8. Regarding the reduction in the number of doctors' visits and the admission of patients in the hospital during the period of Covid19, as well as the lack of reduction in the number of patients in the dialysis department, sufficient explanations should be added.

REPLY: In the first paragraph of the Discussion, lines 400 to 410 in the clean revised manuscript, we summarized our results and remarked that visits to dialysis clinics did not decrease because it is a life-sustaining procedure. Likewise, home care and cancer care declined less than other resources, providing evidence that efforts were made to maintain essential services. We have elaborated upon possible reasons for these and other findings, based on other evidence in the discussion, in lines 428 to 431, and lines 444 to 454 in the clean revised manuscript. 

With respect to the observed decreases in emergency department visits, we explain in lines 428 to 431 that “This is consistent with ED avoidance by people with milder conditions, diversion of human and diagnostic resources away from urgent care, as well as significant decreases in infections associated with lockdown measures.” In lines 444 to 454, the Discussion reads: “In aggregate, these findings demonstrate that the healthcare system was able to continue providing life-sustaining care to patients who needed it, for example those receiving hemodialysis. However, the use of services appeared to decline according to the severity of conditions they are intended to treat, with greater reductions in the use of elective procedures, and preventive, screening, and routine care than urgent and semi-urgent services [35, 80, 81]. Furthermore, after sharp initial declines in use, recovery was slow for many healthcare resources, and their use did not return to pre-pandemic levels by December 2020. Notably, the recovery was much less than needed to manage the backlogs (areas above the curves with negative values, and below the zero-axis), which raises concerns about the residual unmet healthcare needs of the population. This could be a compounding effect if unmet needs resulted in acute events that could have been avoided with earlier care.”

However, our retrospective descriptive study with its inherent study design using administrative data cannot determine causation.

9. The conclusion of the study is one of the main parts. The final conclusion based on the findings of the study should be mentioned - in this section, explain what practical benefits the findings of the present study have for the health system.

REPLY: Thank you for this suggestion. We have revised the Conclusions to propose some practical lessons from the study for the healthcare system. Please see lines 513 to 523 in the clean revised manuscript, which read as follows:

“We observed decreased costs and rates of use for most healthcare resources in two Canadian provinces with predominantly publicly-funded healthcare following the declaration of the COVID-19 pandemic and the associated initial mitigation measures. These changes may have compromised the care of many non-COVID-19 patients in efforts to prepare for the anticipated demand of care for patients with COVID-19. These findings highlight the need for more flexible and sustainable systems that can ensure timely access to adequate care for all individuals. Monitoring to detect and immediately address backlogs could avoid the need for supra-normal levels of activity to "catch up" the system. Further research on the potential consequences of delayed and cancelled medical care will be necessary to mitigate risks in access and quality of healthcare services, address backlog recovery, and avoid healthcare system collapse.”

10. Acknowledgements: a paragraph is enough, it is too long and redundant.

REPLY: Thank you for this comment. These detailed acknowledgements are required verbatim by ICES to acknowledge the providers of all administrative databases used for the Ontario arm of the study.

11. After studying the references: it can be seen that the majority of the references are related to Canadian studies and other studies in other countries have not been used.

REPLY: Thank you for noting this. The healthcare system in Canada provides universal health coverage to its residents and it was important to compare our findings with those from a similar context. Nonetheless, we have added references for recent international studies in the Introduction and the Discussion. Because all of the databases were Canadian, the references for the data (references 37 to 47, and 52 to 55) are from Canadian sources.

12. Very important question: Did the level of communication between doctors and patients, as well as the level of communication between other health personnel and patients, not change during the period of the Covid19 epidemic? (Keeping distance of health personnel from patients, fewer patient visits, fear of covid19 and conflict with this disease) these confounding variables did not have an effect on the amount of use of health services as well as costs???

REPLY: The Reviewer mentions an important point. The outcomes assessed in our study were healthcare costs and resource use following COVID-19 directives and mitigation measures as specified in lines 116-117 of the Introduction, in the clean revised manuscript. We recognize that social distancing, changes in health-seeking behaviour, directives, or patients’ motivation would have influenced the observed changes in our outcomes, but these measures/policies were related to the COVID-19 pandemic, rather than confounding the true effect. 

To address this, we mention in the Introduction (lines 100 to 108), that there were directives to ramp down non-emergency/non-essential healthcare services during the early phases of the COVID-19 pandemic in Canada and other countries. Importantly, in the Strengths and Limitations subsection of the Discussion, in lines 484 to 490, we mention that some of the reductions in healthcare use may have been due to changes to the interpersonal nature of healthcare delivery at the physician level as well as the patient level (e.g. patients’ avoidance of medical settings) and that these may very well have been related to our study outcomes. The Discussion now includes the following sentences on page 26:

“While health system directives were a major factor in the observed changes, some of the reductions in healthcare use may also have been driven by patients’ avoidance of medical settings [80]. However, social distancing behaviours, directives, or patients’ motivation are likely to have acted as mediators, rather than confounders, of the changes in healthcare costs and use that we observed. That is, the behavioural changes and policies are part of the COVID-19 pandemic effect, rather than confounding the true effect.”

Comments from Reviewer #2, and replies: 

1. The manuscript was well presented from title to recommendations, it has clear lines to flow. They have separated the costs’ decline due to the mitigation measures and again the covid related costs, which directly answers the aim of the research. Generally fine with statistical methods used, as the data used was available, starting from descriptive to autocorrelations was typical measure to use in such data, but due to the huge magnitude of the data some errors can be detected (which I didn't focus that much).

REPLY: Thank you for these comments. The reviewer makes a good point about the possibility of errors in such large datasets. The administrative datasets we used are of high quality and they have been extensively used in health services research, as indicated in our response to Reviewer 1, comment 3. The Ontario administrative data from ICES and the data from British Columbia held at Population Data BC are routinely used in studies on healthcare. Please see the references at the following websites that show the use of these administrative health data sources in informing health care and population health in Ontario and BC, respectively: https://www.ices.on.ca/Publications/Journal-Articles and https://www.popdata.bc.ca/ria.

In conducting this study, we considered the following in particular with respect to data accuracy:

We carefully looked over all of the results and performed several validity checks to detect any odd results or inconsistencies within and between the data in Ontario and BC. Moreover, we graphed most of the outcomes for our own benefit to better understand trends and changes; this enabled us to easily observe consistencies between the results on resource use and the results on costs to confirm their robustness.

We excluded people who did not have valid provincial health insurance numbers and performed other checks for valid data. Although the number of people included in the denominator of each weekly analysis varied considerably as people were born, died, moved, and met/did not meet the inclusion/exclusion criteria, there was no evidence of systematic error.

Our study team includes experts in statistics and research methodology who carefully and independently conducted the analyses in BC and Ontario. These individuals have significant experience conducting similar analyses and have collaborated on several studies which used health care administrative data from ICES and Population Data BC. These include the following published papers: 

1. Tsui TCO, Zeitouny S, Bremner KE, Cheung DC, Mulder C, Croxford R, Del Giudice L, Lapointe-Shaw L, Mendlowitz A, Wong WWL, Perlis N, Sander B, Teckle P, Tomlinson G, Walker JD, Malikov K, McGrail KM, Peacock S, Kulkarni GS, Pataky RE, Krahn MD. Initial health care costs for COVID-19 in British Columbia and Ontario, Canada: an interprovincial population-based cohort study. CMAJ Open. 10(3):E818-30. Epub 2022 Sep 20. DOI: https://doi.org/10.9778/cmajo.20210328.

2. McBride ML, de Oliveira C, Duncan R, Bremner KE, Liu N, Greenberg ML, Nathan PC, Rogers PC, Peacock SJ, Krahn MD. Comparing childhood cancer care costs in two Canadian provinces. Healthc Policy. 2020; 15(3):76-88. Epub 2020 Feb 1. DOI: https://doi.org/10.12927/hcpol.2020.26129.

3. de Oliveira C, Pataky R, Bremner KE, Rangrej J, Chan KKW, Cheung WY, Hoch JS, Peacock S, Krahn MD. Estimating the cost of cancer care in British Columbia and Ontario: a Canadian inter-provincial comparison Healthc Policy. 2017; 12(3):95-108.

---

## [Decision Letter · Decision Letter 1]

13 Aug 2023

The impact of the early COVID-19 pandemic on healthcare system resource use and costs in two provinces in Canada: An interrupted time series analysis

PONE-D-23-03072R1

Dear Dr. Cheung,

We’re pleased to inform you that your manuscript has been judged scientifically suitable for publication and will be formally accepted for publication once it meets all outstanding technical requirements.

Kind regards,

Masoud Behzadifar

Academic Editor

PLOS ONE

Additional Editor Comments (optional):

Reviewers' comments:

Reviewer's Responses to Questions

**Comments to the Author**

1. If the authors have adequately addressed your comments raised in a previous round of review and you feel that this manuscript is now acceptable for publication, you may indicate that here to bypass the “Comments to the Author” section, enter your conflict of interest statement in the “Confidential to Editor” section, and submit your "Accept" recommendation.

Reviewer #3: All comments have been addressed

2. Is the manuscript technically sound, and do the data support the conclusions?

Reviewer #3: Yes

3. Has the statistical analysis been performed appropriately and rigorously? 

Reviewer #3: Yes

4. Have the authors made all data underlying the findings in their manuscript fully available?

Reviewer #3: Yes

5. Is the manuscript presented in an intelligible fashion and written in standard English?

Reviewer #3: Yes

6. Review Comments to the Author

Reviewer #3: Title was interesting, which was explained in SMART way.

Introduction part; Clearly mentions magnitude, severity, gaps and solution for identified gap.

Method section; study setting and source of data were clearly mentioned,and also the analysis were wonderful for the study design you used and also mentioned in relevant way.

My Comments is the Acknowledgements were too long and redundant. And it did not clearly mention Data Collection procedures.

7. PLOS authors have the option to publish the peer review history of their article (what does this mean?). If published, this will include your full peer review and any attached files.

Reviewer #3: **Yes: **Belayneh Jejaw Abate

---

## [Editor Report · Acceptance letter]

29 Aug 2023

PONE-D-23-03072R1 

The impact of the early COVID-19 pandemic on healthcare system resource use and costs in two provinces in Canada: An interrupted time series analysis 

Dear Dr. Cheung:

I'm pleased to inform you that your manuscript has been deemed suitable for publication in PLOS ONE. Congratulations! Your manuscript is now with our production department. 

Kind regards, 

on behalf of

Dr. Masoud Behzadifar 

Academic Editor

PLOS ONE